



# First Observations of the McMurdo-South Pole Ionospheric HF Channel

Alex T. Chartier[1], Juha Vierinen[2] and Geonhwa Jee[3]

[1] Johns Hopkins University Applied Physics Laboratory, USA

5   [2] University of Tromsø, Norway

[3] Korea Polar Research Institute, Korea

*Correspondence to:* Alex T. Chartier (alex.chartier@jhuapl.edu)

**Abstract.** We present the first observations from a new low-cost oblique ionosonde located in Antarctica.
10   The transmitter is located at McMurdo Station, Ross Island and the receiver at Amundsen-Scott Station,
South Pole. The system was demonstrated successfully in March 2019, with the experiment yielding over
30 000 ionospheric echoes over a two-week period. These data indicate the presence of a stable E-layer and
a sporadic and variable F-layer with dramatic spread-F of sometimes more than 500 km (in units of virtual
height). The most important ionospheric parameter, NmF2, validates well against the Jang Bogo VIPIR
ionosonde (observing more than 1000 km away). GPS-derived TEC data from the MIDAS algorithm can
be considered necessary but insufficient to predict 7.2 MHz propagation between McMurdo and South Pole,
yielding a true positive in 40% of cases and a true negative in 73% of cases. The success of this pilot
experiment at a total grant cost of $116k and an equipment cost of ~$15k indicates that a large multi-static
network could be built to provide unprecedented observational coverage of the Antarctic ionosphere.


## 1. Introduction

### 1.1 High-latitude ionospheric variability

The high latitude ionosphere frequently exhibits dramatic variability. Some of the first observations of these
phenomena were made by Meek (1949) using an HF sounder at Baker Lake, Canada. Using an
unprecedented and currently unmatched network of ionosondes, Hill (1963) gave the first clear picture of
the phenomenon we have come to understand as the tongue of ionization (e.g. Foster, 1989), and showed
it breaking into a patch. This F-layer ionospheric variability is caused primarily by dense, photo-ionized
plasma being convected into the polar caps (e.g. Lockwood and Carlson, 1992). Most theories explaining
this behavior are skewed heavily towards the northern hemisphere due to better observational coverage
there. Investigations covering the southern hemisphere continue to produce apparently contradictory results.
For example, Noja et al. (2013), Xiong et al. (2018) and Chartier et al. (2019) find more variability around
December/January, whereas Coley and Heelis (1998), Spicher et al. (2018) and David et al. (2019) show a
maximum in June/July. One thing these authors agree on is that the Antarctic ionosphere is far more
variable than the Arctic, up to 2x more variable in summer. New observations are needed in the southern
polar cap to resolve this controversy. The F-layer peak density (called NmF2) must be observed separate
from the E-layer, whose peak density (called NmE) can be equivalent or even greater than NmF2 at high
latitudes (e.g. Hatton, 1961). The horizontal extent of these features can be hundreds or thousands of
kilometers, so a relatively low-cost approach is required that can provide spatially distributed observations.
High temporal cadence is also essential, given that horizontal drift velocities up to 5000 km/hour have been
reported (e.g. Hill, 1963).

### 1.2 Ionospheric remote sensing using radio signals





Radio signal propagation has been integrally linked with ionospheric research since Marconi's famous transatlantic experiment in 1901. The first ionosonde was built by Breit and Tuve (1925). The instrument works by transmitting radio signals of increasing frequency and then receiving their ionospheric echoes. The time-of-flight between transmission and reception is used to estimate their virtual range (by assuming that the signals travelled at the speed of light in free space). The Maximum Observed Frequency ($MOF$) is the
highest frequency signal that is received on the ground. If sufficiently close frequency spacing is used for the transmissions, and if the signal's angle of incidence with the ionosphere ($\vartheta$) is known, $MOF$ can be related to the critical frequency of the ionosphere ($foF2$) by Eq. (1):

$$foF2 \approx MOF \cdot \cos\theta, \tag{1}$$

Once obtained, $foF2$ (in Hertz) is easily converted to $NmF2$ (in electrons/m$^3$) via Eq. (2):

$$NmF2 = \left(\frac{foF2}{9}\right)^2 \tag{2}$$

The same approach can be employed for echoes returned below the peak height, so that bottomside electron density profiles can be retrieved from vertical or oblique ionosondes. Although their numbers have reduced
since the International Geophysical Year (1957), substantial networks of ionosondes exist today, notably the Digisonde network of about 50 instruments (Reinisch et al., 2018) and the sophisticated Vertical Incidence Pulsed Ionospheric Radar (VIPIR) system (Bullett et al., 2016). However, coverage is very sparse in the southern polar cap, with only the VIPIR at Jang Bogo producing reliable, high temporal cadence observations.

1.3 Low-cost, open-source ionospheric remote sensing
The number of ionosondes in existence and the availability of their data are restricted by their typically high cost and proprietary status. Recent developments in meteor radar observation provide a means of solving this problem. Vierinen et al. (2015) observed meteor echoes in Germany using coded continuous
wave transmissions at a fixed frequency, using a software-defined radio system. In communications terms, this is analogous to direct-sequence spread-spectrum modulation. The authors estimate a ~14dB signal processing gain compared to a typical pulsed system, which allows for reduced peak transmitted power. The coded continuous wave approach also reduces false positive detections and allows for a multi-static network of transmitters and receivers to be developed. Signals from different transmitters can be separated through
post-processing because each one uses a different pseudo-random code on the same frequency. Although the technique requires modification for ionospheric remote sensing, its availability through MIT Haystack's DigitalRF software-defined radio package is a major advantage to this investigation.

**2. Method**

2.1 Coded continuous wave ionosonde
We modify the Vierinen et al. (2015) meteor radar approach for ionospheric sounding by adding a frequency-hopping capability. This new code makes the transmitter and receiver step through a pre-defined list of
frequencies at specified seconds past each minute. GPS timing signals trigger the oscillators to retune at precisely the same time in both stations. In the present application, this retuning occurs every five seconds, allowing the system to cover 12 frequencies each minute, but up to 60 frequencies could be used without modification of the underlying software. The frequency schedule can be changed simply by editing text files in the transmitter and receiver computers. The transmitter and receiver bandwidth is effectively 50 kHz
(with 10x oversampling followed by integration and decimation employed at the receiver to reduce noise).



The code consists of pseudo-random binary phase modulations of 1000 bauds in length, yielding 6000 km unaliased range resolution. As with the frequency schedule, these settings may be changed without altering the underlying software. Received signals are autocorrelated with the pseudo-random code to identify the time-of-flight, Doppler shift and intensity of the transmissions. Signals above 6dB (the noise floor of the receiver) are sent back whenever internet access is available, while the raw I/Q is stored on-site for future retrieval and analysis. The result is a remotely-controllable instrument that has a data budget of only a few MB/day and delivers ionospheric soundings at a cadence of one minute. The code for this system is publicly available at github.com/alexchartier/sounder.

### 2.2 Installation in Antarctica

Having received a grant of \$116k from the National Science Foundation's Office for Polar Programs, we developed, built, tested and deployed the system in Antarctica. The system's configuration is shown in Fig. (1). The transmitter is located at McMurdo Station, on the southern exposure of Observation Hill on the former site of the nuclear power station. The receiver is at South Pole Station, with the electronics housed in the V8 vault around 30' under the ice. This configuration provides for oblique sounding of the ionosphere approximately halfway between McMurdo and South Pole.

### 2.2.1 Transmitter

The transmit antenna is a broadband 180' Barker and Williamson tilted, terminated folded dipole costing around \$2000 and mounted in an east-west inverted-vee configuration on a 50' central mast and 15' stub masts. The transmitter electronics are made up of an Ettus Research USRP N210 with GPS-disciplined oscillator and BasicTx daughterboard. The final-stage amplifier is a Motorola-designed AN762-180 producing <50W effective transmitted power. Over the course of the experiment, the amplifier developed distortion leading to excessive Standing-Wave Ratio (SWR) and so is not recommended for future installations.

### 2.2.2 Receiver

At the receiver site, an inexpensive 1m active broadband dipole antenna is mounted around 8' above the ice and connected to the receiver by 600' of RG-6 cable. The received signal is boosted ~20dB by a low-noise amplifier and connected to a USRP N210 with BasicRx daughterboard and GPS-disciplined oscillator. The total equipment cost is approximately \$15k. Both sites have internet connection (typically 8/24 hours at South Pole) so observations are typically returned within a day of being taken. The system has been reconfigured to use different frequencies and changed output power levels at various stages.

### 2.3 Data processing

Ionospheric products are estimated by selecting the shortest range returns at the highest frequencies in the E- and F-region virtual height intervals (60-180 and 180-600 km). The shortest range return at a given frequency is selected because it represents the signal that has the smallest azimuthal deviation from great-circle propagation. The signal's angle of incidence with the ionosphere is estimated following Eq. (3):

$$\vartheta = C \sin^{-1} (\Delta_{MCM\_ZSP} / R) \tag{3}$$

where $\Delta_{MCM\_ZSP}$ is the distance between McMurdo and South Pole (1356 km), R is the observed range and C is a calibration factor used to account for a reduction in the angle of incidence due to signal refraction. Based on empirical comparison with the Jang Bogo VIPIR data, we use a calibration factor of 0.9 in the E-region and 0.75 in the F-region.

### 2.4 Validation data





Data from the VIPIR system in operation at the Korean Antarctic station Jang Bogo [*Bullett et al., 2016*] are used for validation. The VIPIR system uses 4000W transmitted power, a sophisticated log-periodic transmit antenna and Dynasonde data processing. There is approximately 1000 km separation between the observing areas of the two instruments, so the comparison with VIPIR is not expected to be exactly one-to-one. However, ground-based GPS-derived Total Electron Content observations are available co-located with our new system. We use TEC images produced using the Multi-Instrument Data Analysis Software (MIDAS) algorithm (Mitchell and Spencer, 2003; Spencer and Mitchell, 2007) at a 15-minute cadence. The algorithm solves for electron densities in a nonlinear, three-dimensional, time-dependent algorithm based on dual-frequency GPS phase data. These images are interpolated to the midpoint between South Pole and McMurdo (83.93 S, 166.69 E) to provide a first-order comparison against the data from our RF experiment. Note that a single pixel of the TEC images extends about 500 km horizontally, so the exact reflection location is not critical to this comparison.

## 3. Results

### 3.1 Results of the McMurdo-South Pole demonstration

The system was operated between 28 February and 13 March at 12 frequencies between 2.6 and 7.2 MHz. These are listed in Table 1. No signals were received below 4.1 MHz, due to absorption and reduced transmitter efficiency. No signal was received on 4.4 MHz for unknown reasons. Histograms of intensity, virtual height and Doppler velocity of the working frequencies are shown in Figure 2. The observed ranges and Doppler velocities are tightly clustered within physically realistic parts of the system's unaliased observing scope, which covers more than 2500 km of virtual height and 3000 m/s of Doppler velocity. The virtual heights show two distributions, with E- and F-layer echoes clearly separated on frequencies up to 6 MHz and only F-layer echoes ($>200$ km) at 7.2 MHz. The observed Doppler velocities are typically small with a small negative bias. The local time distribution of the echoes shows a clear peak between 15-21 LT on all frequencies, consistent with the expectation that sporadic-F should occur in the local afternoon/evening. The local time distribution may explain the negative Doppler bias, given that the F-region tends to move upwards during this time interval.

A total of 30543 ionospheric echoes were received. Of the working channels, the largest number of echoes was received on 5.1 MHz, and the least on 6.4 MHz. The number of echoes received on 5.1 MHz (21517) is actually 25% higher than the number of minutes in the test period (17280) because echoes are frequently received at multiple ranges, from both the E- and F-layers at the same time. This multi-mode propagation is possible because the signal's angle of incidence is different for the two layers (larger for the E-layer) and because the signal scatters. Multi-mode propagation can be seen clearly in virtual height-time-intensity data shown in Figure 3, especially on 5.1 MHz. The E-layer is clearly visible on 4.1, 5.1 and 6.0 MHz, with stable virtual height of 100-120 km. The F-region echoes, by contrast, exhibit sporadic variability on the higher frequencies. Some of these sporadic-F enhancements are spread in virtual height by 500+ km, most notably on 5.1 MHz.

### 3.2 Validation against Jang Bogo VIPIR

NmF2 from the McMurdo-South Pole experiment is compared against that observed by the Jang Bogo VIPIR in Figure 4. The diurnal variability of NmF2 is consistent across both datasets, though the oblique experiment observes a smaller range of values due to its lower frequency resolution. The Jang Bogo VIPIR collects more data due to its higher sensitivity.



### 3.3 Comparison with ground-based GPS TEC

MIDAS TEC data at the reflection point are compared against the maximum frequency (7.2 MHz) HF returns to determine whether density enhancements are correlated across the two datasets. Results are shown in Figure 5. Considering TEC at the midpoint between McMurdo and South Pole as a predictor of 7.2 MHz links between the two stations, TEC > 6 TECU predicts propagation successfully 40% of the time, and TEC < 6 TECU predicts absence of propagation 73% of the time.


## 4. Discussion

Data from the experiment indicate that the system performed successfully. Virtual height and Doppler distributions are physically realistic, indicating that the transmitter and receiver clocks and oscillators were synchronized correctly. The virtual height distributions show a clear separation between E- and F-layer echoes. The system's observing capability (>2500 km virtual height and 3000 m/s Doppler) vastly exceeds the physically-expected range of values and the reported observations lie inside the expected regions of the system. All these factors show that the reported echoes are real ionospheric reflections originating from
McMurdo and received at South Pole. The NmF2 observations validate well against the Jang Bogo VIPIR data.

Propagation on 7.2 MHz was predicted moderately well by MIDAS GPS TEC (40% true positive, 73% true negative), given the inherent differences between the two datasets. This indicates that relatively high
observed TEC can be considered a necessary but insufficient condition for predicting 7.2 MHz propagation between McMurdo and South Pole. Variability beyond the spatio-temporal resolution of the available GPS TEC data may explain the disparity between the two datasets. Such mesoscale variability is known to be higher during times of enhanced F-region density, when steep ionospheric density gradients and high velocities are common.


Enormous spreading of virtual height is frequently observed. Echoes are received simultaneously across intervals of over 500 km virtual height. If the signal is instead being spread horizontally, this data could be interpreted as indicating azimuthal deviations of up to 35 degrees off of great circle propagation (assuming a 200-km reflection height). This is on the order of the azimuthal spreads reported by *Bust et al.* [1994] and
*Flaherty et al.* [1996] at middle and low latitudes.

## 5. Summary

A new oblique ionosonde has been developed and installed at McMurdo and South Pole stations. A two-week system demonstration yielded over 30 000 ionospheric echoes indicating a stable E-layer and a sporadic and variable F-layer with dramatic range spreading. The experiment validates well against the Jang Bogo VIPIR vertical ionosonde and MIDAS GPS-derived TEC data. Given the success of this pilot experiment and the low cost of the equipment (~$15k), this technology could be used to build a large network of
ionospheric sounders operating multi-statically to provide new scientific insights into high-latitude ionospheric behavior.





## 6. Acknowledgements

Thanks to all the people who helped with the technical developments of this work, notably Nate Temple (Ettus Research), Ryan Volz (MIT Haystack), Ethan Miller (then APL), Ben Witvliet (University of Bath), Owen Crise (then Maret School), Mike Legatt, Liz Widen & all from Antarctic Support Contract, Andrew Knuth (APL), The RF Connection of Gaithersburg MD, Virgil Stamps of HF Projects, Andy Gerrard, Gareth Perry and Hyomin Kim of New Jersey Institute of Technology. The authors thank Cathryn Mitchell of the University of Bath, U.K., for providing access to the Multi-Instrument Data Analysis System (MIDAS). Thanks to the International GNSS Service for the GPS data. The raw GPS data used in this paper are available at ftp://garner.ucsd.edu, ftp://geodesy.noaa.gov and ftp://data-out.unavco.org. The precise orbit files are also available at ftp://garner.ucsd.edu. The sounder code developed for this work is available at https://github.com/alexchartier/sounder. The data used to make all the figures in this paper are available at https://zenodo.org/record/3478267 under DOI 10.5281/zenodo.3478267.

ATC acknowledges support from National Science Foundation grants OPP-1643773, AGS-1341885 and AGS-1934973. GJ acknowledges support from PE19020 that funds the VIPIR observations at JBS.

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





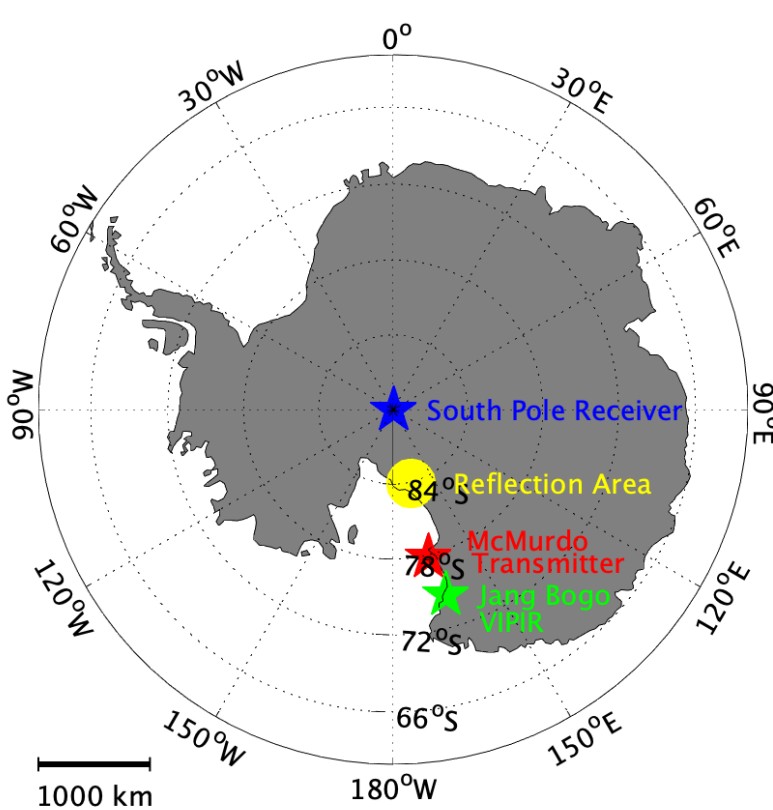

Figure 1: Experimental configuration with transmitter at McMurdo (red), receiver at South Pole (blue), approximate ionospheric reflection area (for single hop propagation) at midpoint between them (yellow) and validation instrument (VIPIR) at Jang Bogo station (green).








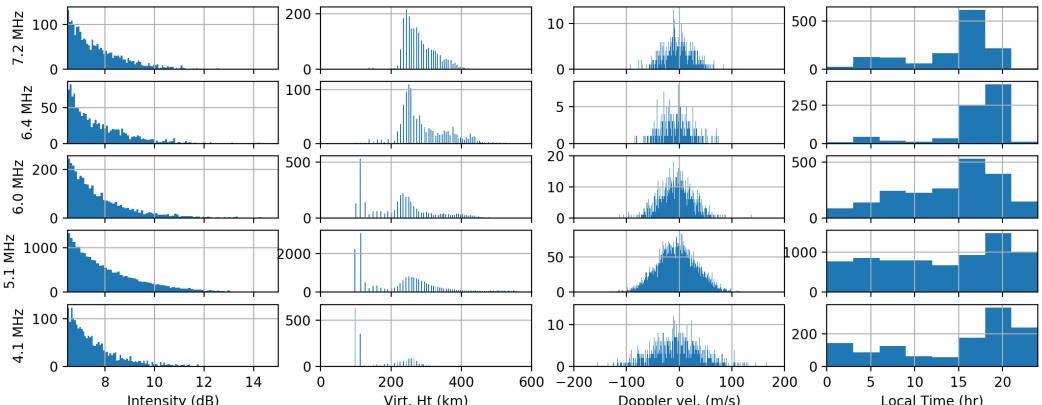

Figure 2 shows histograms of received intensity, virtual height, Doppler velocity (positive for decreasing path lengths) and local time of received echoes.

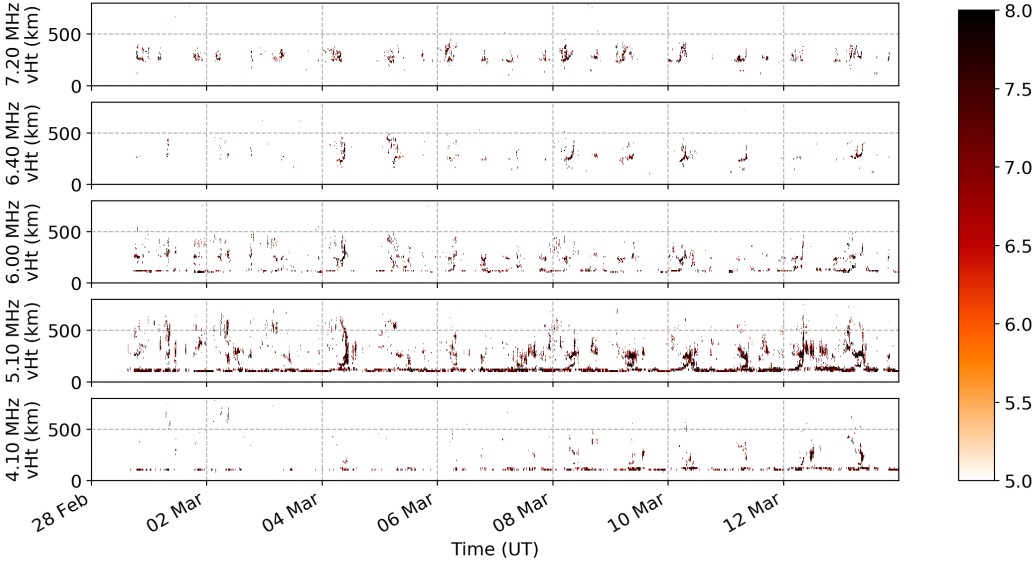

Figure 3 shows virtual height-time-intensity data from the technology demonstration experiment (transmitter at McMurdo, receiver at South Pole). The E-layer is consistently visible at 100-120 km on 4.1 and 5.1 MHz. Sporadic F-region enhancements are seen around local noon (UT + 12) on the higher frequencies. These are accompanied by dramatic virtual height spreading of 500km+.






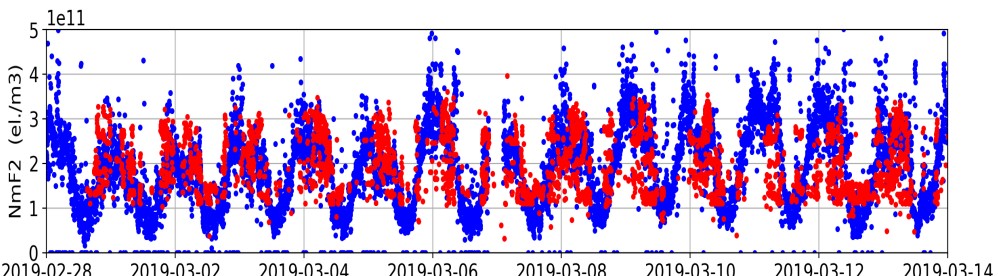

Figure 4 shows NmF2 from the Jang Bogo VIPIR (red) and McMurdo-South Pole oblique experiment (blue). The two datasets are consistent, but the McMurdo-South Pole experiment shows a smaller range of values because it uses fewer frequencies.

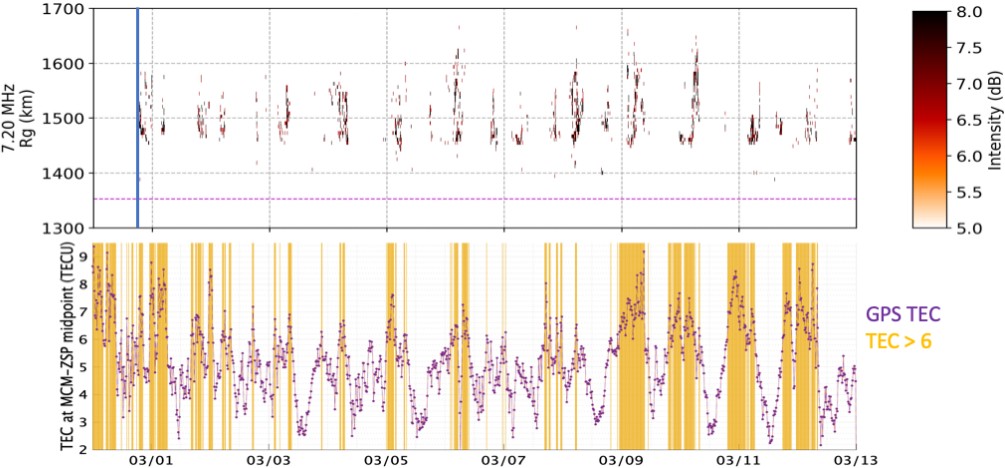

Figure 5 shows (above) 7.2 MHz echoes from the McMurdo-South Pole demonstration and (below) MIDAS GPS TEC at the midpoint between McMurdo and South Pole. TEC values > 6 TECU are highlighted to illustrate the correspondence between TEC enhancements and sporadic F-region propagation. The time when the transmitter was switched on is shown on the upper plot in blue.


Table 1: # of ionospheric echoes received between 28 February and 15 March.

| Frequency (MHz) | # of echoes received |
|---|---|
| 7.2 | 2234 |
| 6.4 | 1189 |
| 6.0 | 3474 |





| | |
|---|---|
| **5.1** | 21517 |
| **4.4** | 0 |
| **4.1** | 2129 |
| **3.7** | 0 |
| **3.4** | 0 |
| **3.2** | 0 |
| **3.0** | 0 |
| **2.8** | 0 |
| **2.6** | 0 |



