# Peer review of "First Observations of the McMurdo-South Pole Oblique Ionospheric HF Channel"

_Atmospheric Measurement Techniques, 2020_

## Referee Comment (RC1) · Anonymous Referee #1 · 27 Feb 2020

Summary
* * *
In this manuscript high frequency (HF) ionospheric sounder utilizing modern software defined radio (SDR) techniques is described. The sounder was deployment in Antarctica to provide ionospheric diagnostics in an effort to provide more data coverage for studying the polar cap ionosphere. In contrast to sounders on satellites, this sounder composed of a bi-static ground based system with a single antenna transmitter site nearby the McMurdo Station and the receiver site located at South Pole.

First results from the sounder are presented and compared with data from the VIPIR ionosonde and MIDAS TEC maps. The comparisons show largely good agreement

despite the broad geographical distance between the sounder volume and VIPIR and the significant mismatch in spatial resolution between the sounder and MIDAS. The comparisons seem to validate that the HF sounder is functioning as expected.

While I find the scientific significance of the paper to be excellent, both the scientific quality and the presentation quality require improvement. For example, there are missing references, confused references, and some misuse of terminology that confuses the message of the paper. There is also a need to include some of the theoretical details that enable oblique sounding to function as it does as well as are required for understanding and interpreting the data from an oblique sounder. The instrument is not as well explained as it could be and inclusion of more details with theoretical background information would improve the scientific quality of the paper. For these reasons, I recommend the manuscript be accepted subject to major revision.

Major Comments

————

Below follows selected major comments. These are structural and/or comments about clarity of ideas and arguments. These are crucial and need to be addressed.

1) Incorrect reference? The citation of Vierinen et al. [2015] near line 70 appears to be incorrect. I believe the authors meant to cite this 2016 publication instead: https://www.atmos-meas-tech.net/9/829/2016/ Please fix this/clarify.

2) Missing citations! A significant number of the citations in the paper are missing from the references section!

a. Vierinen et al. (2015)

b. Vierinen et al. (2016)

c. Lockwood and Carlson (1992)

d. Noja et al. (2013)

e. Chartier et al. (2019)

f. Coley and Heelis (1998)

g. Spicher et al. (2018)

h. David et al. (2019)

i. Breit and Tuve (1925)

Also, I believe that the authors need to cite the Digital RF project, as requested by the developers: https://github.com/MITHaystack/digital_rf#Citation There is an acknowledgement of the use of Digital RF in the acknowledgement section, but a citation needs to be added as well.

3) In section 1.2, the text here requires at least some basic discussion about magnetoionic theory. Fundamentally, the propagation of radio waves through the ionosphere is described by the Appleton-Hartree equation. Fundamentally, this is why the equation relating foF2 and MOF is approximate, because it assumes some things such as spatial uniformity of the vertical ionospheric plasma density profile, which might not hold true. A brief couple of sentences about where equation 1 comes from, the fundamental physical principles underlying it, and the assumptions baked in are needed here. Similarly, equation 2 needs to be explained to be an approximation of the equation relating plasma frequency and electron density. One should also note that this equation is only valid for the ordinary propagation mode, whereas the extraordinary propagation mode includes a gyrofrequency dependence. For oblique propagation nearly perpendicular to the magnetic field (known as Quasi-transverse propagation), there is still mode splitting. Does the new sounder account for mode splitting (via polarization measurements, assuming the splitting is negligible and if so that should be discussed here)?

4) Near line 70, there is mention of a "∼14 dB signal processing gain", but there is absolutely no context for this claim. The closest thing I can find is this sentence directly in Vierenen et al. (2016) [https://www.atmos-meas-tech.net/9/829/2016/]:

[Figure]

"For example, a continuous transmission would result in ∼14 dB of increased signal processing gain when compared to a pulsed system with a duty cycle of 4.4 %"

But this isn't actually what is commonly understood as a signal processing gain. The 14 dB gain results from increasing the duty cycle from 4.4% to 100% ($10 \log(1/0.044) = $ ∼13.6 dB). Of course there will also be a processing gain associated with matched filtering to the pseudo random code, but this gain isn't discussed anywhere in the current manuscript. The lack of context and confusing terminology needs to be addressed, which could be as simple as expanding the discussion to note that increasing the duty cycle provide an effective gain over pulsed systems in addition to the gain provided by coding of the transmission. Processing gain is typically the terminology used in communications, whereas pulse compression is typically the terminology used in radar. Both are the result of using spread spectrum techniques (i.e. effectively what the pulse coding does), not changing duty cycle.

5) Near line 95: "Signals above 6dB": Is this relative to some absolute or relative power measurement? At HF, the noise environment is known to be highly variable with time of day. Please clarify and expand on this.

6) Either in the "Method" section, or in the "Data processing" section, there needs to be some discussion of how the sounder works: such as how time of flight between the Tx and Rx sites is used to infer the virtual height. Likely this could be done near line 130 in the discussion of equation 3. Only a sentence or two is needed. As it is now, there is a lot of inference required from the reader to understand how this works.

Minor Comments/Corrections

‾‾‾‾‾‾‾

Below are selected minor corrections, largely composed of grammar and spelling corrections. Some may be stylistic and can be treated as suggestions.

1) Near line 40, "5000 km/hour": please meters per second

2) "2x" near line 35 and "10x" near line 90, write these out as "2 times" and "10 times"

3) Written differently, equation 2 is actually an equation for the electron density in terms of the plasma frequency, where all the constants have been approximated by 9. As such, it would be better to either:

a. Rewrite this equation using the full equation for plasma frequency, or

b. Use the approximately equal symbol, instead of the equals symbol.

4) Near line 60, please define "high temporal cadence"? This could be done with a time in brackets, such as ($\sim$5 minutes). For example, the CADI ionosondes in Canada produce and ionogram once every $\sim$5 minutes.

5) Near line 60, it might be useful to compare the number of ionosondes in 1957 to the 7 ionosondes maintained by the Canadian High Arctic Ionospheric Network, which are located in the Canadian Arctic (see: http://chain.physics.unb.ca/chain/pages/data_availability)

6) Near line 70: "The number of ionosondes in existence and the availability of their data are restricted by their typically high cost and proprietary status." How much does an ionosonde typically cost? Can a reference be provided?

7) Near line 75: "Signals from different transmitters can be separated through post-processing because each one uses a different pseudo-random code on the same frequency." Some discussion about how this works, or a citation would be beneficial. Some readers will not be familiar with how phase coding and matched filtering techniques work.

8) Near line 90: "pseudo-random binary phase modulations of 1000 bauds": It might be clearer to also state the baud length (20 us). This makes it easier to see how one obtains 6000 km unambiguous range.

9) Near line 115, does the "effective transmitted power" mean the RF power leaving the amplifier? This terminology sounds similar to "effective radiated power" which combines antenna gain and RF power into the antenna. Please clarify. Sorry to hear that the amplifier degraded like it did!

10) Near line 120, is the LNA attached to the receive antenna or is it a pre-amp to between the N210 and RG-6?

11) Near line 120: suggested "The system has been remotely reconfigured to use different frequencies and changed output power levels at various stages."

12) Near line 120: Since this is a new instrument, it might be beneficial to explain how the data is collected and processed. Voltage samples are saved using DigitalRF? and then post processed how? Here might be a good place to refer readers to specific equations or sections of Vierinen et al. 2016 for parts of the processing that is identical.

13) Near line 130: Are there any plans to model the calibration factor C? One should be able to estimate the factor with an inverse problem where the forward model predicts the time of flight by ray tracing through a model ionosphere. A good candidate model ionosphere that works at high latitude might be E-CHAIM (doi: 10.1002/2017JA024398). At the very least, such a model could provide an apriori from which a perturbation electron density profile could be inferred from the measured time of flight compared to the modeled time of flight.

14) Near line 160: "which covers more than >2500 km of virtual height and 3000 m/s Doppler velocity". Is this 3000 m/s capability +/- or total? All of this could be discussed together in one section/subsection where a full description of the new sounder is given.

---

## Referee Comment (RC2) · Anonymous Referee #2 · 30 Mar 2020

Referee #2

This manuscript presents demonstration oblique ionosonde observations from the Antarctic. The ionosonde is of a new low-cost design and employs advanced modern digital hardware and software defined radio signal waveform and signal processing techniques. Observations from the demonstration campaign in March 2019 were verified with a traditional ionosonde reasonably closely co-located. Propagation of the 7.2 MHz oblique ionosonde channel is compared with GPS-derived TEC observations using the MIDAS model to access if density enhancements and propagation are correlated between the two observation systems, giving moderately good agreement.

The scientific importance of this manuscript is of great interest and in need of publication as it presents a novel concept for probing the ionosphere in greater detail than

previously. However, the manuscript needs improvement. As many of my comments and concerns with the manuscript are essentially verbatim with those of Referee #1, there is no point in repeating them here. As I strongly agree with Referee #1, I will only include additional comments and suggestions and try not to repeat things.

In general, more details are needed on the new oblique ionosonde demonstration instrument and how a network, specifically a multi-static network, in the Antarctic will benefit ionospheric research.

Specific Comments

lines 1-2, title: I would suggest highlighting the oblique ionosonde aspect in the title, as ionosondes typically operate in both the MF and HF bands (however, it is recognized that no MF data was available in this demonstration experiment due to technical issues).

Section 1.1: Are there only scientific questions of interest in the Antarctic ionosphere dealing with its variability? A few other examples of the new abilities and questions which could be answered with an oblique ionosonde network in the Antarctic is needed? Contrast benefits/challenges associated with oblique versus vertical observations, etc.

Section 2.1: Much more detail on the new oblique ionosonde is needed. For example: 1) the unaliased range resolution is given, but this needs to be related to (virtual) height measurements; 2) Doppler resolution is not given, although the Doppler extent is given but at a much later point in the manuscript; 3) what is the range-gate size?; 4) what is the baud length?; 5) is there time averaging and, if so, what is it and how does this relate to the 5-seconds between frequency switches?; 6) how were the frequencies selected for this study? 7) why not use 60 frequencies for a sweep if the instrument was capable of this as stated?; and so on. A succinct and convenient method to present this instrument technical data, or at least most of it, is in a table. It makes for easy comparison to other instruments.

lines 133-134: Please include a description of the methodology used to produce the calibration factors, C_E and C_F, for the E- and F-regions. Please justify the calibration factors due to its importance relating virtual range to virtual height.

line 139: Include a reference to Dynasonde data processing if not already supplied in Bullett et al., 2016. Also, present key parameters of the VIPIR Jang Bogo ionosonde and compare to the new oblique ionosonde. If this new instrument is to complement current ionosonde networks, how it compares to them is of great interest.

Section 3.1, line 167: Is it possible to show an oblique ionogram from the new ionosonde? However, it is understandable that these ionograms may not 'look' like a typical ionogram due to the lack of sweep frequencies – only 12 were available and only 5 of those received signal.

Section 3.3 and lines 203-209 in Discussion section: I am not sure of the point of the comparison with ground-based TEC measurements and MIDAS. What is unique about TEC being greater than or less than 6 TECU and how does this related to 7.2 MHz? And how/why was 7.2 MHz selected? What is the expected outcome of this comparison?

line 225: Again, how does the "multi-static" configuration of a large network of oblique ionosondes supply new insights into the ionosphere? What would be the benefit of this?

Technical Comments

line 41: Please include, in parenthesis, the standard notation used to express drift velocity values in the ionosphere.

lines 53 and 56 (referring to Equations 1 and 2): Reference(s) is needed for equations. Equ. 2 is well know, but still should be referenced; Equ. 1 is not so well know, at least at this time.

Section 1.3: Suggest last sentence (lines 75-77) should come after sentence on line

71. A reference is needed for Digital RF.

line 105: What is a 'V8 vault'? Reference. And/or short description. What was the transmitter equipment housed in?

lines 111-112: Please include references for N210 and Motorola AN762-180 transmitter.

line 159: Virtual height and maximum Doppler velocity are parameters which should have been first presented in Section 2.1. Is a virtual height of 2500 km scientifically useful?

line 182: The VIPIR ionosonde does have higher sensitivity, but is not the reason it collects more data compared to the oblique ionosonde mostly due to the fact that fewer sweep frequencies were used by the oblique ionosonde? This is noted in the caption for Figure 4, but not in the main text.
* * *

---

## Author Response (AR1)

**AMT Response to Reviewer RC-1**

1) Incorrect reference
**Changed**

2) Missing citations
**Added**

3) In section 1.2, the text here requires at least some basic discussion about magnetoionic theory.

**We have added the following: "It is important to note that the formulation in Eqs. (1) and (2) is based only on ordinary-mode wave propagation and that mode splitting can occur in the presence of transverse magnetic fields, further adding to the uncertainty of *NmF2* retrievals. Budden (1961) provides a comprehensive description of radio wave propagation in the ionosphere."**

4) Near line 70, there is mention of a "~14 dB signal processing gain"

**We have removed that number to avoid confusion.**

5) Near line 95: "Signals above 6dB":

**Changed to: "All signals >6dB above the noise floor of the receiver"**

6) Either in the "Method" section, or in the "Data processing" section, there needs to be some discussion of how the sounder works: such as how time of flight between the Tx and Rx sites is used to infer the virtual height. Likely this could be done near line 130 in the discussion of equation 3

**Added the following explanation: "The virtual height is calculated from the range assuming simple triangular raypath geometry with a single reflection at the midpoint between McMurdo and South Pole. Earth's radius at the midpoint (required to calculate the height of the reflection) is taken from the World Geodetic System 1984 model (WGS84). Note that virtual height is larger than true height because it assumes propagation at the speed of light in free space and ignores signal refraction near the reflection point."**

**Note that we have made all the code available so interested readers may see how we have implemented these and other calculations. This one occurs in calc_dist() and calc_vht() within plot_rtd.py.**

Minor:

1) Near line 40, "5000 km/hour": please meters per second

**We prefer to keep the figure and units as stated in the original paper than to introduce a conversion that could potentially be erroneous (at least they are metric…)**

2) "2x" near line 35 and "10x" near line 90, write these out as "2 times" and "10 times"

**changed**

3) Written differently, equation 2 is actually an equation for the electron density in terms of the plasma frequency, where all the constants have been approximated by 9. As such, it would be better to either: a. Rewrite this equation using the full equation for plasma frequency, or b. Use the approximately equal symbol, instead of the equals symbol.

**Changed (approx. equal sign used)**

4) Near line 60, please define "high temporal cadence"? This could be done with a time in brackets, such as (~5 minutes). For example, the CADI ionosondes in Canada produce and ionogram once every ~5 minutes.

**Added (2-min). Incidentally, 5 minutes might not be enough to catch a supposed typical "patch" (e.g. 200-km diameter travelling at 1 km/s is well within the bounds of the literature, and would pass over a point location in less than four minutes).**

5) Near line 60, it might be useful to compare the number of ionosondes in 1957 to the 7 ionosondes maintained by the Canadian High Arctic Ionospheric Network, which are located in the Canadian Arctic (see: http://chain.physics.unb.ca/chain/pages/data_availability)

**Added**

6) Near line 70: "The number of ionosondes in existence and the availability of their data are restricted by their typically high cost and proprietary status." How much does an ionosonde typically cost? Can a reference be provided?

**Unfortunately we do not have a citation available to include a specific figure in the manuscript, but $100 000 - $200 000 is typical.**

7) Near line 75: "Signals from different transmitters can be separated through postprocessing because each one uses a different pseudo-random code on the same frequency." Some discussion about how this works, or a citation would be beneficial. Some readers will not be familiar with how phase coding and matched filtering techniques work.

**Added a reference to Vierinen et al. (2016) where this information came from.**

8) Near line 90: "pseudo-random binary phase modulations of 1000 bauds": It might be clearer to also state the baud length (20 us). This makes it easier to see how one obtains 6000 km unambiguous range.

**Added**

9) Near line 115, does the "effective transmitted power" mean the RF power leaving the amplifier? This terminology sounds similar to "effective radiated power" which combines antenna gain and RF power into the antenna. Please clarify. Sorry to hear that the amplifier degraded like it did!

**The description has been updated: "Based on the power/SWR meter installed on-site, we estimate that the system produced <50W total transmitted power. Over the course of the experiment, the amplifier developed distortion leading to excessive Standing-Wave Ratio (SWR) and out-of-band emissions, so it is not recommended for future installations."**

**Our power meter showed figures between 30 – 100 W and SWR of 1.5 – 3 (both frequency-dependent), so we estimate that the power leaving the antenna never exceeded 50W (50% of power is reflected back towards the amplifier when SWR = 3).**

10) Near line 120, is the LNA attached to the receive antenna or is it a pre-amp to between the N210 and RG-6?

**The LNA is downstream of the bias tee, inside the vault. New text reads: "At the receiver site, an inexpensive 1m active broadband dipole antenna is mounted around 8' above the ice and connected to the receiver by 600' of RG-6 cable. The antenna receives phantom power from a bias tee located in the vault. Inside the vault, the signal is boosted ~20dB by a low-noise amplifier and connected to a USRP N210 with BasicRx daughterboard and GPS-disciplined oscillator."**

11) Near line 120: suggested "The system has been remotely reconfigured to use different frequencies and changed output power levels at various stages."

**Changed to: "At various stages during testing, the system was remotely reconfigured through secure shell connection (SSH) to use different frequencies and output power levels."**

12) Near line 120: Since this is a new instrument, it might be beneficial to explain how the data is collected and processed. Voltage samples are saved using DigitalRF? and then post processed how? Here might be a good place to refer readers to specific equations or sections of Vierinen et al. 2016 for parts of the processing that is identical.

**Section 2.1 contains most of this information already, but we have added some clarifications to that part. The section now reads:**

**"We modify the Vierinen et al. (2016) meteor radar approach for ionospheric sounding by adding a frequency-hopping capability. This new code makes the transmitter and receiver step through a pre-defined list of frequencies at specified seconds past each minute. GPS timing signals trigger the oscillators to retune at precisely the same time in both stations. In the present application, this retuning occurs every five seconds, allowing the system to cover 12 frequencies each minute, but up to 60 frequencies could be used without modification of the underlying software. The frequency schedule can be changed simply by editing text files in the transmitter and receiver computers. Aside from that modification, the system is essentially unchanged from that used by Vierinen et al. (2016). The transmitter and receiver bandwidth is effectively 50 kHz (with 10 times oversampling followed by integration and decimation employed at the receiver to reduce noise). The code consists of pseudo-random binary phase modulations of 1000 bauds in length, each 20 μs duration, yielding 6000 km unaliased range resolution. Received signals have DC offsets removed, have non-Gaussian components rejected to mitigate radio-frequency interference, and are then autocorrelated with the pseudo-random code to produce range-Doppler-intensity matrices for each analysis period (5-s). All signals >6dB above the noise floor of the receiver are sent back as sparse matrices whenever internet access is available, while the raw I/Q is stored on-site in DigitalRF format for future retrieval and analysis. The result is a remotely-controllable instrument that has a data budget of only a few MB/day and delivers ionospheric soundings at a cadence of one minute. The code for this system is publicly available at github.com/alexchartier/sounder."**

13) Near line 130: Are there any plans to model the calibration factor C? One should be able to estimate the factor with an inverse problem where the forward model predicts the time of flight by ray tracing through a model ionosphere. A good candidate model ionosphere that works at high latitude might be E-CHAIM (doi: 10.1002/2017JA024398). At the very least, such a model could provide an apriori from which a perturbation electron density profile could be inferred from the measured time of flight compared to the modeled time of flight.

**We are working on raytracing approaches to this and other HF datasets, and will keep E-CHAIM in mind.**

14) Near line 160: "which covers more than >2500 km of virtual height and 3000 m/s Doppler velocity". Is this 3000 m/s capability +/- or total? All of this could be discussed together in one section/subsection where a full description of the new sounder is given.

**The text has been modified to clarify that this is total resolution: "(>2500 km virtual height and 3000 m/s total Doppler resolution)." The physically-expected Doppler is very small because the system is observing apparent vertical motion for the most part. The ExB component (observed for example by SuperDARN) is much larger than the vertical component, yet even that is typically below 1000 m/s. We make this point about resolution here in the discussion simply to point out that the system appears to be working correctly.**

*In general, more details are needed on the new oblique ionosonde demonstration instrument and how a network, specifically a multi-static network, in the Antarctic will benefit ionospheric research.*

*Specific Comments*

*lines 1-2, title: I would suggest highlighting the oblique ionosonde aspect in the title, as ionosondes typically operate in both the MF and HF bands (however, it is recognized that no MF data was available in this demonstration experiment due to technical issues).*

We have added the word oblique to the title: "First Observations of the McMurdo-South Pole Oblique Ionospheric HF Channel"

*Section 1.1: Are there only scientific questions of interest in the Antarctic ionosphere dealing with its variability?*

We have added a note on the question of E-F coupling and the scientific potential of the Antarctic ionosphere:

"Another area of current scientific interest is E-F coupling, where forcing applied to the E-layer through neutral dynamics or other drivers appears to map into the F-layer (e.g. Cosgrove and Tsunoda, 2004; Saito et al., 2007). This phenomenon is perhaps easiest to observe at high Magnetic latitudes, where the dip angle is almost vertical and so any E-F coupling should be spatially localized, rather than being separated by hundreds or thousands of kilometers as is the case at middle or low latitudes. In general, the Antarctic ionosphere is of great scientific interest because it provides potentially the best ground base from which to observe deep polar cap dynamics, which may reveal new insights into direct coupling between the Solar wind and Earth's atmosphere. This is because Magnetic fieldlines at very high latitudes are typically "open" rather than closing in the magnetosphere."

*A few other examples of the new abilities and questions which could be answered with an oblique ionosonde network in the Antarctic is needed? Contrast benefits/challenges associated with oblique versus vertical observations, etc.*

The following text has been added to 1.2:
"Oblique sounding has some advantages compared to vertical-mode operation for ionospheric sounding. Principal among these is the ability to observe a location (or locations) in the ionosphere spatially separated from the ground infrastructure. This is important when operating in remote areas such as Antarctica, where the cost of installing and maintaining ground stations

is high. Of course, this benefit comes with an associated challenge in interpreting the data, as the signal path through the ionosphere is unknown. Oblique sensing also provides for potentially large networks of observations to be built using a relatively small number of transmitters. That is important because HF transmitters require far more power and larger antennas than receivers, and also often create broadcast licensing issues. Therefore, oblique sounding may be useful in expanding the spatial coverage of ionospheric observations, especially in remote areas."

*Section 2.1: Much more detail on the new oblique ionosonde is needed. For example: 1) the unaliased range resolution is given, but this needs to be related to (virtual) height measurements; 2) Doppler resolution is not given, although the Doppler extent is given but at a much later point in the manuscript; 3) what is the range-gate size?; 4) what is the baud length?; 5) is there time averaging and, if so, what is it and how does this relate to the 5-seconds between frequency switches?; 6) how were the frequencies selected for this study? 7) why not use 60 frequencies for a sweep if the instrument was capable of this as stated?; and so on. A succinct and convenient method to present this instrument technical data, or at least most of it, is in a table. It makes for easy comparison to other instruments.*

We have added a table (now Table 1) covering all relevant instrument parameters. The choice to use 12 frequencies is now explained: "the smaller number of frequencies allows for longer integration time and therefore increases sensitivity." There is no time averaging beyond the 5-second integration period.

*lines 133-134: Please include a description of the methodology used to produce the calibration factors, C_E and C_F, for the E- and F-regions. Please justify the calibration factors due to its importance relating virtual range to virtual height.*

An explanation for the calibration factors has been added: "C is a calibration factor used to account for a reduction in the angle of incidence due to signal refraction. Based on empirical comparison with the Jang Bogo VIPIR data, we use a calibration factor of 0.9 in the E-region and 0.75 in the F-region."

*line 139: Include a reference to Dynasonde data processing if not already supplied in Bullett et al., 2016. Also, present key parameters of the VIPIR Jang Bogo ionosonde and compare to the new oblique ionosonde. If this new instrument is to complement current ionosonde networks, how it compares to them is of great interest.*

References added. Figure 4 provides a direct comparison of the most reliable ionosonde parameter, NmF2, from VIPIR Jang Bogo and our new instrument.

*Section 3.1, line 167: Is it possible to show an oblique ionogram from the new ionosonde? However, it is understandable that these ionograms may not 'look' like a typical ionogram due to the lack of sweep frequencies – only 12 were available and only 5 of those received signal.*

As the reviewer notes, ionogram-style plots do not add that much information as they have a maximum of five points on them. However, we have added all the daily range-time-intensity and range-time-Doppler plots as supporting information, so interested readers can see what the underlying data looks like.

*Section 3.3 and lines 203-209 in Discussion section: I am not sure of the point of the comparison with ground-based TEC measurements and MIDAS. What is unique about TEC being greater than or less than 6 TECU and how does this related to 7.2 MHz? And how/why was 7.2 MHz selected? What is the expected outcome of this comparison?*

The following explanation has been added to the text: "MIDAS TEC data at the reflection point are compared against the maximum frequency (7.2 MHz) HF returns in order to determine whether observed density enhancements are correlated across the two datasets. High TEC values at the midpoint between McMurdo and South Pole could be a predictor of maximum-frequency (7.2 MHz) links between the two stations because of the association between NmF2 (and therefore critical frequency) and TEC. A 6 TECU threshold was found to provide a good association with the 7.2 MHz propagation data. Several free parameters escape this comparison, notably variations in signal absorption from the D- and E-layers, scattering by irregularities, variations in the peak height (hmF2) and sub-grid-scale variability missed by the TEC images, so an exact match is not expected."

*line 225: Again, how does the "multi-static" configuration of a large network of oblique ionosondes supply new insights into the ionosphere? What would be the benefit of this?*

An explanation has been added: "Such a network would dramatically expand the spatial coverage of ionospheric observations while requiring a relatively small number of new transmitters."

*Technical Comments*

*line 41: Please include, in parenthesis, the standard notation used to express drift velocity values in the ionosphere.*

Added: "High temporal cadence is also essential, given that horizontal drift velocities of 5000 km/hour (approximately 1400 m/s) have been reported by Hill (1963)."

*lines 53 and 56 (referring to Equations 1 and 2): Reference(s) is needed for equations. Equ. 2 is well know, but still should be referenced; Equ. 1 is not so well know, at least at this time.*

Equation 1 is an approximation of the well-known relation FoF2 = MUF/sec theta. A reference has been added to Budden (1961), who provides a thorough description of ionospheric HF propagation.

*Section 1.3: Suggest last sentence (lines 75-77) should come after sentence on line 71. A reference is needed for Digital RF.*

Text reordered (see also response to other reviewer). Volz et al (2019) added.

*line 105: What is a 'V8 vault'? Reference. And/or short description. What was the transmitter equipment housed in?*

It is "the" V8 Vault: a plywood enclosure that was buried just below the ice some years ago and has sunk down over the years, with sections of ladder added periodically. The description has been expanded as follows: "The receiver is at South Pole Station, with the electronics housed in the V8 Vault (also home to VLF electronics used by LaBelle et al., 2015), currently located around 30' under the ice around 1 km from the main base"

The transmitter was housed in a galvanized steel shed manufactured by Northern Tool and assembled by me at McMurdo. Apparently it survived quite well (see Figure 1 attached).

[Figure]

*lines 111-112: Please include references for N210 and Motorola AN762-180 transmitter.*

These have been added.

*line 159: Virtual height and maximum Doppler velocity are parameters which should have been first presented in Section 2.1. Is a virtual height of 2500 km scientifically useful?*

These have been added to Section 2.1. The scientific utility of large observing scope is explained there: "The observed ranges and Doppler velocities are tightly clustered within physically realistic parts of the system's unaliased observing scope,"

If the observing scope were much smaller, it would not be obvious that the system is working properly. As it is, the probability of our data landing at random in the physically-realistic part of the observing scope is small, so we believe the system is working as expected.

*line 182: The VIPIR ionosonde does have higher sensitivity, but is not the reason it collects more data compared to the oblique ionosonde mostly due to the fact that fewer sweep frequencies were used by the oblique ionosonde? This is noted in the caption for Figure 4, but not in the main text.*

The statement has been modified: "The Jang Bogo VIPIR reports more NmF2 values due to its higher sensitivity, and due to the fact that it covers more frequencies."

**First Observations of the McMurdo-South Pole Oblique Ionospheric HF Channel**

Alex T. Chartier[1], Juha Vierinen[2] and Geonhwa Jee[3]

[1] Johns Hopkins University Applied Physics Laboratory, USA

[2] University of Tromsø, Norway

[3] Korea Polar Research Institute, Korea

*Correspondence to:* Alex T. Chartier (alex.chartier@jhuapl.edu)

**Abstract.** We present the first observations from a new low-cost oblique ionosonde located in Antarctica. The transmitter is located at McMurdo Station, Ross Island and the receiver at Amundsen-Scott Station, South Pole. The system was demonstrated successfully in March 2019, with the experiment yielding over 30 000 ionospheric echoes over a two-week period. These data indicate the presence of a stable E-layer and a sporadic and variable F-layer with dramatic spread-F of sometimes more than 500 km (in units of virtual height). The most important ionospheric parameter, NmF2, validates well against the Jang Bogo VIPIR ionosonde (observing more than 1000 km away). GPS-derived TEC data from the MIDAS algorithm can be considered necessary but insufficient to predict 7.2 MHz propagation between McMurdo and South Pole, yielding a true positive in 40% of cases and a true negative in 73% of cases. The success of this pilot experiment at a total grant cost of $116k and an equipment cost of ~$15k indicates that a large multi-static network could be built to provide unprecedented observational coverage of the Antarctic ionosphere.

**1. Introduction**

**1.1 High-latitude ionospheric variability**

The high latitude ionosphere frequently exhibits dramatic variability. Some of the first observations of these phenomena were made by Meek (1949) using an HF sounder at Baker Lake, Canada. Using an unprecedented and currently unmatched network of ionosondes, Hill (1963) gave the first clear picture of the phenomenon we have come to understand as the tongue of ionization (e.g. Foster, 1989), and showed it breaking into a patch. This F-layer ionospheric variability is caused primarily by dense, photo-ionized plasma being convected into the polar caps (e.g. Lockwood and Carlson, 1992). Most theories explaining this behavior are skewed heavily towards the northern hemisphere due to better observational coverage there. Investigations covering the southern hemisphere continue to produce apparently contradictory results. For example, Noja et al. (2013), Xiong et al. (2018) and Chartier et al. (2019) find more variability around December/January, whereas Coley and Heelis (1998), Spicher et al. (2017) and David et al. (2019) show a maximum in June/July. One thing these authors agree on is that the Antarctic ionosphere is far more variable than the Arctic, up to twice as variable during summer. New observations are needed in the southern polar cap to resolve this controversy. The F-layer peak density (called NmF2) must be observed separate from the E-layer, whose peak density (called NmE) can be equivalent or even greater than NmF2 at high latitudes (e.g. Hatton, 1961). The horizontal extent of these features can be hundreds or thousands of kilometers, so a relatively low-cost approach is required that can provide spatially distributed observations. High temporal cadence is also essential, given that horizontal drift velocities of 5000 km/hour (approximately 1400 m/s) have been reported by Hill (1963). Another area of current scientific interest is E-F coupling, where forcing applied to the E-layer through neutral dynamics or other drivers appears to map into the F-layer (e.g. Cosgrove and Tsunoda, 2004; Saito et al., 2007). This phenomenon is perhaps easiest to observe at high Magnetic latitudes, where the dip angle is almost vertical and so any E-F coupling

should be spatially localized, rather than being separated by hundreds or thousands of kilometers as is the case at middle or low latitudes. In general, the Antarctic ionosphere is of great scientific interest because it provides potentially the best ground base from which to observe deep polar cap dynamics, which may reveal new insights into direct coupling between the Solar wind and Earth's atmosphere. This is because Magnetic fieldlines at very high latitudes are typically "open" rather than closing in the magnetosphere.

1.2 Ionospheric remote sensing using radio signals

Radio signal propagation has been integrally linked with ionospheric research since Marconi's famous transatlantic experiment in 1901. The first ionosonde was built by Breit and Tuve (1925). The instrument works by transmitting radio signals of increasing frequency and then receiving their ionospheric echoes. The time-of-flight between transmission and reception is used to estimate their virtual range (by assuming that the signals travelled at the speed of light in free space). The Maximum Observed Frequency ($MOF$) is the highest frequency signal that is received on the ground. If sufficiently close frequency spacing is used for the transmissions, and if the signal's angle of incidence with the ionosphere ($\vartheta$) is known, $MOF$ can be related to the critical frequency of the ionosphere ($foF2$) by Eq. (1):

$$foF2 \approx MOF \cdot \cos\theta, \tag{1}$$

Once obtained, $foF2$ (in Hertz) is easily converted to $NmF2$ (in electrons/m$^3$) via Eq. (2):

$$NmF2 \approx \left(\frac{foF2}{9}\right)^2 \tag{2}$$

The same approach can be employed for echoes returned below the peak height, so that bottomside electron density profiles can be retrieved from vertical or oblique ionosondes. It is important to note that the formulation in Eqs. (1) and (2) is based only on ordinary-mode wave propagation and that mode splitting can occur in the presence of transverse magnetic fields, further adding to the uncertainty of $NmF2$ retrievals. Budden (1961) provides a comprehensive description of radio wave propagation in the ionosphere.

Although their numbers have reduced since the International Geophysical Year (1957), substantial networks of ionosondes exist today, notably the Digisonde network of about 50 instruments (Reinisch et al., 2018), the Canadian High Arctic Ionosonde Network of about 7 instruments, and several installations of the sophisticated Vertical Incidence Pulsed Ionospheric Radar (VIPIR) system (Bullett et al., 2016). However, coverage is very sparse in the southern polar cap, with only the VIPIR at Jang Bogo producing reliable, high (2-min) temporal cadence observations.

Oblique sounding has some advantages compared to vertical-mode operation for ionospheric sounding. Principal among these is the ability to observe a location (or locations) in the ionosphere spatially separated from the ground infrastructure. This is important when operating in remote areas such as Antarctica, where the cost of installing and maintaining ground stations is high. Of course, this benefit comes with an associated challenge in interpreting the data, as the signal path through the ionosphere is unknown. Oblique sensing also provides for potentially large networks of observations to be built using a relatively small number of transmitters. That is important because HF transmitters require far more power and larger antennas than receivers, and also often create broadcast licensing issues. Therefore, oblique sounding may be useful in expanding the spatial coverage of ionospheric observations, especially in remote areas.

1.3 Low-cost, open-source ionospheric remote sensing

The number of ionosondes in existence and the availability of their data are restricted by their typically high cost and proprietary status. Recent developments in meteor radar observation provide a means of solving this problem. Vierinen et al. (2016) observed meteor echoes in Germany using coded continuous wave transmissions at a fixed frequency, using a software-defined radio system. In communications terms, this is analogous to direct-sequence spread-spectrum modulation. The coded continuous wave approach provides substantial signal processing gain and allows for reduced peak transmitted power compared to a pulsed system. It also reduces false positive detections and allows for a multi-static network of transmitters and receivers to be developed. Signals from different transmitters can be separated through post-processing because each one uses a different pseudo-random code on the same frequency (Vierinen et al., 2016). Although the technique requires modification for ionospheric remote sensing, its availability through MIT Haystack's DigitalRF software-defined radio package (Volz et al., 2019) is a major advantage to this investigation. A separate open-source ionospheric sounder (also based on a meteor radar technique) has recently been developed by *Bostan et al.* (2019) and is available as part of the GnuRadar package.

**2. Method**

**2.1 Coded continuous wave ionosonde**

We modify the Vierinen et al. (2016) meteor radar approach for ionospheric sounding by adding a frequency-hopping capability. This new code makes the transmitter and receiver step through a pre-defined list of frequencies at specified seconds past each minute. GPS timing signals trigger the oscillators to retune at precisely the same time in both stations. In the present application, this retuning occurs every five seconds, allowing the system to cover 12 frequencies each minute, but up to 60 frequencies could be used without modification of the underlying software – the smaller number of frequencies allows for longer integration time and therefore increases sensitivity. Frequencies were selected based on IRI raytracing (with a larger extent selected than was indicated by the raytracing) and to remain >250 kHz away from any Antarctic HF communications channel. We coordinated with McMurdo Comms to test for any interference with their operations and found no effect on their system. However, in principle the hardware supports operation anywhere in the HF band, while the software could operate at any frequency, and the operating schedule can be changed simply by editing text files in the transmitter and receiver computers. Aside from that modification, the system is essentially unchanged from that used by Vierinen et al. (2016). The transmitter and receiver bandwidth is effectively 50 kHz (with 10 times oversampling followed by integration and decimation employed at the receiver to reduce noise). The code consists of pseudo-random binary phase modulations of 1000 bauds in length, each 20 μs duration, yielding 6000 km unaliased range resolution. Table 1 provides a summary of the instrument characteristics as configured for this experiment.

The following data processing is applied to the received 50-kHz baseband signal centered on each frequency. Signals have DC offsets removed, have non-Gaussian components rejected to mitigate radio-frequency interference, and are then autocorrelated with the pseudo-random code to produce range-Doppler-intensity matrices for each 5-s analysis period. All signals >6dB above the noise floor of the receiver are sent back as sparse range-Doppler-intensity matrices whenever internet access is available, while the raw I/Q is stored on-site in DigitalRF format for future retrieval and analysis. The result is a remotely-controllable instrument that has a data budget of only a few MB/day and delivers ionospheric soundings at a cadence of one minute. The code for this system is publicly available at github.com/alexchartier/sounder.

**2.2 Installation in Antarctica**

Having received a grant of $116k from the National Science Foundation's Office for Polar Programs, we developed, built, tested and deployed the system in Antarctica. The total equipment cost for this experiment was approximately $15k. The system's configuration is shown in Fig. (1). The transmitter is located at McMurdo Station, on the southern exposure of Observation Hill on the former site of the nuclear power station, with the electronics housed in a galvanized steel cabinet assembled by the PI. The receiver is at South Pole Station, with the electronics housed in the V8 Vault (also home to VLF electronics used by LaBelle et al., 2015), currently located around 30' under the ice ~1 km from the main base. This configuration provides for oblique sounding of the ionosphere approximately halfway between McMurdo and South Pole. Both sites have internet connection (typically 8/24 hours at South Pole) so observations are typically returned to Lab servers for analysis within a day of being taken. At various stages during testing, the system was remotely reconfigured through secure shell connection (SSH) to use different frequencies and output power levels.

2.2.1 Transmitter

The transmit antenna is a broadband 180' Barker and Williamson tilted, terminated folded dipole costing around $2000 and mounted in an east-west inverted-vee configuration on a 50' central mast and 15' stub masts. The transmitter electronics are made up of an Ettus Research USRP N210 with GPS-disciplined oscillator and BasicTx daughterboard (National Instruments, 2020). The final-stage amplifier is a Motorola-designed AN762-180 (Granberg, 1976). Based on the power/SWR meter installed on-site, we estimate that the system produced <50W total transmitted power. Over the course of the experiment, the amplifier developed distortion leading to excessive Standing-Wave Ratio (SWR) and out-of-band emissions, so it is not recommended for future installations.

2.2.2 Receiver

At the receiver site, an inexpensive 1m active broadband dipole antenna is mounted around 8' above the ice and connected to the receiver by 600' of RG-6 cable. The antenna receives phantom power from a bias tee located in the vault. Inside the vault, the signal is boosted ~20dB by a low-noise amplifier and connected to a USRP N210 with BasicRx daughterboard and GPS-disciplined oscillator. The receiver's oscillator retunes according to the pre-defined frequency schedule based on a timed command triggered by the GPS PPS signal.

2.3 Data processing

Ionospheric products are estimated by selecting the shortest range returns at the highest frequencies in the E- and F-region virtual height intervals (60-180 and 180-600 km). The shortest range return at a given frequency is selected because it represents the signal that has the smallest azimuthal deviation from great-circle propagation. The signal's angle of incidence with the ionosphere is estimated following Eq. (3):

$$\vartheta = C \sin^{-1}\left(\Delta_{MCM\_ZSP} / R\right) \qquad (3)$$

where $\Delta_{MCM\_ZSP}$ is the distance between McMurdo and South Pole (1356 km), R is the observed range and C is a calibration factor used to account for a reduction in the angle of incidence due to signal refraction. Based on empirical comparison with the Jang Bogo VIPIR data, we use a calibration factor of 0.9 in the E-region and 0.75 in the F-region. The virtual height is calculated from the range assuming simple triangular raypath geometry with a single reflection at the midpoint between McMurdo and South Pole. Earth's radius at the midpoint (required to calculate the height of the reflection) is taken from the World Geodetic System 1984 model (WGS84). Note that virtual height is larger than true height because it assumes propagation at the speed of light in free space and ignores signal refraction near the reflection point.

**2.4 Validation data**

Data from the VIPIR system in operation at the Korean Antarctic station Jang Bogo (Bullett et al., 2016; Kwon et al., 2018) are used for validation. The VIPIR system uses 4000W transmitted power, a sophisticated log-periodic transmit antenna and Dynasonde data processing, described by Zabotin et al. (2016). There is approximately 1000 km separation between the observing areas of the two instruments, so the comparison with VIPIR is not expected to be exactly one-to-one. However, ground-based GPS-derived Total Electron Content observations are available co-located with our new system. We use TEC images produced using the Multi-Instrument Data Analysis Software (MIDAS) algorithm (Mitchell and Spencer, 2003; Spencer and Mitchell, 2007) at a 15-minute cadence. The algorithm solves for electron densities in a nonlinear, three-dimensional, time-dependent algorithm based on dual-frequency GPS phase data. These images are interpolated to the midpoint between South Pole and McMurdo (83.93 S, 166.69 E) to provide a first-order comparison against the data from our RF experiment. Note that a single pixel of the TEC images extends about 500 km horizontally, so the exact reflection location is not critical to this comparison.

**3. Results**

**3.1 Results of the McMurdo-South Pole demonstration**

The system was operated between 28 February and 13 March at 12 frequencies between 2.6 and 7.2 MHz. These are listed in Table 2. No signals were received below 4.1 MHz, due to absorption and reduced transmitter efficiency. No signal was received on 4.4 MHz for unknown reasons. Histograms of intensity, virtual height and Doppler velocity of the working frequencies are shown in Figure 2. The observed ranges and Doppler velocities are tightly clustered within physically realistic parts of the system's unaliased observing scope, which covers more than 2500 km of virtual height and 3000 m/s of Doppler velocity. The virtual heights show two distributions, with E- and F-layer echoes clearly separated on frequencies up to 6 MHz and only F-layer echoes (>200 km) at 7.2 MHz. The observed Doppler velocities are typically small with a small negative bias. The local time distribution of the echoes shows a clear peak between 15-21 LT on all frequencies, consistent with the expectation that sporadic-F should occur in the local afternoon/evening. The local time distribution may explain the negative Doppler bias, given that the F-region tends to move upwards during this time interval.

A total of 30543 ionospheric echoes were received. Of the working channels, the largest number of echoes was received on 5.1 MHz, and the least on 6.4 MHz. The number of echoes received on 5.1 MHz (21517) is actually 25% higher than the number of minutes in the test period (17280) because echoes are frequently received at multiple ranges, from both the E- and F-layers at the same time. This multi-mode propagation is possible because the signal's angle of incidence is different for the two layers (larger for the E-layer) and because the signal scatters. Multi-mode propagation can be seen clearly in virtual height-time-intensity data shown in Figure 3, especially on 5.1 MHz. The E-layer is clearly visible on 4.1, 5.1 and 6.0 MHz, with stable virtual height of 100-120 km. The F-region echoes, by contrast, exhibit sporadic variability on the higher frequencies. Some of these sporadic-F enhancements are spread in virtual height by 500+ km, most notably on 5.1 MHz.

**3.2 Validation against Jang Bogo VIPIR**

NmF2 from the McMurdo-South Pole experiment is compared against that observed by the Jang Bogo VIPIR in Figure 4. The diurnal variability of NmF2 is consistent across both datasets, though the oblique

270 experiment observes a smaller range of values due to its lower frequency resolution. The Jang Bogo VIPIR reports more NmF2 values due to its higher sensitivity, and due to the fact that it covers more frequencies.

**3.3 Comparison with ground-based GPS TEC**

275 MIDAS TEC data at the reflection point are compared against the maximum frequency (7.2 MHz) HF returns in order to determine whether observed density enhancements are correlated across the two datasets. High TEC values at the midpoint between McMurdo and South Pole could be a predictor of maximum-frequency (7.2 MHz) links between the two stations because of the association between NmF2 (and therefore critical frequency) and TEC. A 6 TECU threshold was found to provide a good association with the 7.2 MHz propagation data. Several free parameters escape this comparison, notably variations in signal
280 absorption from the D- and E-layers, scattering by irregularities, variations in the peak height (hmF2) and sub-grid-scale variability missed by the TEC images, so an exact match is not expected. Results are shown in Figure 5. TEC > 6 TECU predicts propagation successfully 40% of the time, and TEC < 6 TECU predicts absence of propagation 73% of the time.

285

**4. Discussion**

[revised manuscript text omitted]

Figure 2 shows histograms of received intensity, virtual height, Doppler velocity (positive for decreasing path lengths) and local time of received echoes.

[Figure]

Figure 3 shows virtual height-time-intensity data from the technology demonstration experiment (transmitter at McMurdo, receiver at South Pole). The E-layer is consistently visible at 100-120 km on 4.1 and 5.1 MHz. Sporadic F-region enhancements are seen around local noon (UT + 12) on the higher frequencies. These are accompanied by dramatic virtual height spreading of 500km+.

470

[Figure]

Figure 4 shows NmF2 from the McMurdo-South Pole oblique experiment (red) and Jang Bogo VIPIR (blue). The two datasets are consistent, but the McMurdo-South Pole experiment shows a smaller range of values because it uses fewer frequencies.

[Figure]

Figure 5 shows (above) 7.2 MHz echoes from the McMurdo-South Pole demonstration and (below) MIDAS GPS TEC at the midpoint between McMurdo and South Pole. TEC values > 6 TECU are highlighted to illustrate the correspondence between TEC enhancements and sporadic F-region propagation. The time when the transmitter was switched on is shown on the upper plot in blue.

475

Table 1: Instrument characteristics

| Instrument parameter | Value |
|---|---|
| Bandwidth | 50 kHz (10x oversampled, followed by integration and decimation) |
| Gaussian phase encoding | 1000 bauds, each 20 µs in length |
| Frequency hopping | 12 frequencies (see Table 2) each minute, 5-second dwell |
| Range | 6000 @ 6 km res. |
| Virtual height | 2885 @ <15 km (E-layer), <7 km (F-layer) |

| | |
|---|---|
| Doppler | 2885 @ 11.5 m/s (for 2.6 MHz) down to 1042 @ 4.2 (for 7.2 MHz) |
| Integration period | 5-seconds (using 12 frequencies each minute |
| Data budget | 6.31 TB/year raw IQ (50 kHz sc16), approx. 1 GB/year retrieved parameters |
| Power budget | Approx. 150 W at the transmitter, 30 W at the receiver |

480

Table 2: Number of ionospheric echoes received between 28 February and 15 March.

| Frequency (MHz) | # of echoes received |
|---|---|
| 7.2 | 2234 |
| 6.4 | 1189 |
| 6.0 | 3474 |
| 5.1 | 21517 |
| 4.4 | 0 |
| 4.1 | 2129 |
| 3.7 | 0 |
| 3.4 | 0 |
| 3.2 | 0 |
| 3.0 | 0 |
| 2.8 | 0 |
| 2.6 | 0 |

485

490

495

500